# An Adaptive IMM Algorithm for a PD Radar with Improved Maneuvering Target Tracking Performance

**Wenwen Xu, Jiankang Xiao, Dalong Xu, Hao Wang \*** and **Jianyin Cao**

School of Electronic and Optical Engineering, Nanjing University of Science and Technology, Nanjing 210094, China; wenwenxu@njust.edu.cn (W.X.); xudl1987@njust.edu.cn (D.X.); jianyin.cao@njust.edu.cn (J.C.)

\* Correspondence: haowang@njust.edu.cn

**Abstract:** A pulse-Doppler (PD) radar has the advantage of strong anti-interference ability, and it is often used as a solution for maneuvering target tracking. In the application of target monitoring and tracking in PD radars, the interacting multiple model algorithm (IMM) has become the main and preferred choice due to its flexibility and high accuracy. However, the probability transfer matrix in classical IMM algorithms generally depends on constant prior knowledge, and if a PD radar is tracking a strong maneuvering target, it is inevitable to encounter some limitations, such as the possibility of target tracking trajectory deviation, and even a loss of the target. The Markov probability transfer matrix is proposed with an adaptive modification ability in real time to overcome the above problems in this paper. Additionally, for improving the speed of switching between the models, the fuzzy control system for secondary updating of model probability is adopted. By this means, the tracking accuracy of maneuvering targets is enhanced. Compared with the classical IMM algorithm, the corresponding simulation results for the PD radar indicate that the overall tracking accuracy of the proposed adaptive IMM algorithm is improved by 19.6%. In conclusion, the continuity and accuracy of the target trajectory can be effectively improved with the proposed adaptive IMM algorithm in PD radar cases.

**Keywords:** target tracking; interacting multiple model; probability transfer matrix; PD radar

## 1. Introduction

In the past 20 years, the rapid development of electronic communication technology in civil aviation, national defense, and military industries has increased the maneuverability of consumer-level unmanned aerial vehicles (UAVs) and military reconnaissance aircrafts [1–3]. Significant challenges and difficulties have been posed for pulse-Doppler (PD) radars by technological advancements and the increasing complexity of target environments. In order to adapt to the emerging trend of enhanced maneuverability in the target domain, it is necessary to enhance the tracking performance of PD radar systems when dealing with highly maneuverable targets. The interacting multiple model (IMM) algorithm has gradually become a mainstream maneuvering target tracking algorithm due to its superior robustness, accuracy, flexibility, and scalability advantages compared with other single-model tracking algorithms [4–8]. As a result, it has been widely applied in various types of PD radars.

The switching between different models in the model set of the IMM algorithm can be considered as a Markov process [9,10]. The probability of the model at the next moment is not related to the probability of the model at the past moment, but only to the probability of the model at the current moment and the probability transfer matrix [11,12]. Due to the important role of the probability transfer matrix in model switching, many scholars have carried out related optimization work on this concept, such as an error compression rate of models is defined as prior information [1], and it is used for the real-time correction of

the probability transfer matrix to improve the tracking accuracy of the IMM algorithm. However, if the maneuverability of the target decreases, it will lead to an increase in tracking errors. A special-structure IMM algorithm for the adaptive correction of probability transfer matrix is proposed (ATPM-PIMM) [3]. An IMM algorithm is added to the ATPM-PIMM algorithm that adaptively modifies the probability transfer matrix to the framework of the classical IMM algorithm, as a parallel algorithm of the classical IMM algorithm. Based on the designed selection strategy and threshold, the classic IMM algorithm and the adaptive IMM algorithm are adaptively selected. The ATPM-PIMM algorithm is adaptive in switching algorithms and modifies the probability transfer matrix in real time, and is easier to implement. Due to the addition of parallel algorithms, the complexity of the ATPM-PIMM algorithm has increased about twofold. Moreover, the setting of the switching threshold in the ATPM-PIMM algorithm depends on reliable experience, and effective testing is required for algorithm practice. In [13], a new IMM algorithm with a model set design method, which utilized model likelihood function information to modify the probability transfer matrix, was designed. But it mainly applied the model sets of two models, which had significant limitations. In conclusion, the above IMM algorithms have their own innovative optimization ideas, but at the same time, there are also some significant limitations.

Therefore, in order to address the limitations of these algorithms, an adaptive IMM algorithm (ATPFC-IMM) is proposed in this paper, which adds an adaptive modified probability transfer matrix module in the single-step iteration of the classical IMM algorithm. In addition, a fuzzy control system is designed for the ATPFC-IMM algorithm to update the model information after the probability is updated in the classical IMM algorithm, thus addressing the aforementioned issues. Through subsequent simulations and applications, it is demonstrated that the proposed ATPFC-IMM algorithm is effective in enhancing the maneuvering target tracking performance, such as the reduction in trajectory errors and improvement in trajectory continuity in PD radars compared with the classical IMM (CIMM) algorithm and ATPM-PIMM [3] algorithm.

The subsequent sections of this paper are organized as follows. Section 2 provides a concise overview of the processing steps involved in the IMM algorithm. Section 3 delves into a comprehensive discussion of the principles and processing techniques of the ATPFC-IMM algorithm, which is proposed in this study. Section 4 mainly includes a simulation and analysis, where two sets of cases are designed to test the superior performance of the ATPFC-IMM algorithm in model switching and position tracking errors. In Section 5, the ATPFC-IMM algorithm is applied to specific PD radars, and the good tracking performance of the ATPFC-IMM algorithm for maneuvering targets is verified.

## 2. Review of the Classical IMM Algorithm

The movement of maneuvering targets is complex and cannot be described by a single-motion model; so, the multiple-model algorithm is often applied to maneuvering target tracking for PD radars. As a special type of multiple-model tracking algorithm, the interactive multiple model (IMM) algorithm has good tracking performance and flexibility. The IMM algorithm considers the switching between motion models to adapt to the motion state of maneuvering targets.

The classical interacting multiple model algorithm consists of the following four main parts: (1) Model interaction; (2) The parallel filtering of models; (3) The updating of model probability; and (4) Model estimation fusion [14]. The IMM algorithm assumes that the true motion model of the target is obtained by summing the individual model sets of the respective weights they occupy [15–17].

A concise overview of the processing procedure for the classical interactive multiple model algorithm (CIMM) is provided in this paper for comparison. It primarily focuses on the state estimation and model probabilities of each model involved in this procedure. The process of this algorithm is summarized and categorized into four main parts, as shown in the following sections.

### 2.1. Model Interaction

First, a Markov probability transfer matrix, $P_t$, is presented based on a priori information or experience in Equation (1). In addition, $P_t$ remains constant throughout the cyclic processing of the CIMM algorithm.

$$P_t = \begin{bmatrix} P_{t_{11}} & P_{t_{12}} & \cdots & P_{t_{1r}} \\ P_{t_{21}} & P_{t_{22}} & \cdots & P_{t_{r2}} \\ \vdots & \vdots & \ddots & \vdots \\ P_{t_{r1}} & P_{t_{r2}} & \cdots & P_{t_{rr}} \end{bmatrix} \tag{1}$$

where $P_{t_{ij}}$ denotes the probability that model $i$ switches to model $j$.

The model set at moment $k-1$ is the model matching the target motion, refer to Equation (2). the mixed-state estimate after the transfer of other models to the model is calculated as $\hat{X}_{0j}^A(k-1|k-1)$; $\mu_{i|j}(k-1|k-1)$ denotes the probability of switching from other models to matching model $i$.

$$\hat{X}_{0j}^A(k-1|k-1) = \sum_{i=1}^{M} \hat{X}_i^A(k-1|k-1)\mu_{i|j}(k-1|k-1) \tag{2}$$

$$P_{0j}^A(k-1|k-1) = \sum_{i=1}^{M} \left\{ P_i^A(k-1|k-1) + [\hat{X}_i^A(k-1|k-1) - \hat{X}_{0j}^A(k-1|k-1)] \cdot \\ [\hat{X}_i^A(k-1|k-1) - \hat{X}_{0j}^A(k-1|k-1)]' \right\} \mu_{i|j}(k-1|k-1) \tag{3}$$

$$\begin{cases} \mu_{i|j}(k-1|k-1) = \frac{1}{\bar{c}_j} P_{t_{ij}} \mu_i(k-1) \\ \bar{c}_j = \sum_{i=1}^{M} P_{t_{ij}} \mu_i(k-1) \end{cases} \tag{4}$$

### 2.2. Parallel Filtering of Models

The Kalman filtering method [18–21] was used in this part; the state estimate obtained in the previous step was used as the input to model $i$. The state estimate of model $i$ at moment $k$ is $\hat{X}_i(k)$, the covariance matrix is $P_i(k)$, the mean of error is $V_i(k)$, and the matrix $S_i(k)$ (corresponding to the covariance matrix about $V_i(k)$) is obtained after predictive filtering [22–25].

### 2.3. The Updates of Model Probability

Once the models are defined and initialized, the IMM algorithm predicts the future state of each model independently based on their dynamics and system measurements. These predictions are essential for generating reliable estimates and facilitating effective model selection. The probability of model $i$ is calculated from its corresponding model likelihood function $\Lambda_i(k)$:

$$\hat{\mu}_i(k) = \Lambda_i(k)\bar{c}(k)/c(k) \tag{5}$$

$$\Lambda_i(k) = \frac{1}{(2\pi|S_j(k)|)^{1/2}} \exp\left(-\frac{1}{2} V_i(k)^T S_i^{-1}(k) V_i(k)\right) \tag{6}$$

$$c(k) = \sum_{i=1}^{M} \Lambda_i(k)\bar{c}(k) \tag{7}$$

### 2.4. Model Estimation Fusion

The information obtained through the aforementioned three steps can be utilized for state information fusion and serve as the output of the algorithm [26–28]: the state estimation, $\hat{X}(k|k)$, and the covariance of state estimation, $P(k|k)$.

$$\hat{X}(k|k) = \sum_{i=1}^{M} \hat{X}_i(k|k)\mu_i(k) \tag{8}$$

$$P(k|k) = \sum_{i=1}^{M} \mu_i(k)\left\{P_i^A(k|k) + [\hat{X}_i^A(k|k) - \hat{X}(k|k)][\hat{X}_i^A(k|k) - \hat{X}(k|k)]'\right\} \tag{9}$$

As shown from the above steps, the classical IMM (CIMM) algorithm has the advantages of fewer steps and a clear overall processing approach. In order to adapt to the motion transformation of maneuvering targets, multi-model algorithms have been proposed and exhibit an excellent comprehensive performance. Different from traditional multi-model algorithms, the model set of the IMM algorithm can be personalized according to the applications. Moreover, the interaction and switching between models are taken into consideration in this algorithm. Due to the good flexibility and scalability of the IMM algorithm, the IMM algorithm becomes a good candidate for optimizing algorithm structure and adding adaptive modules. Therefore, the proposed ATPFC-IMM algorithm adds the probability transfer matrix adaptive real-time correction and fuzzy control system for re-updating the model's probability on the basis of the classical IMM algorithm's structure.

### 3. The Design of the ATPFC-IMM Algorithm

Due to the fact that the output of the CIMM algorithm comes from the interaction, filtering, updating, and fusion between various models in the model set, the probability transfer matrix plays an important role. And the probability transfer matrix in the CIMM algorithm is mainly based on constant prior experience, even in tracking maneuvering targets. Therefore, the ATPFC-IMM algorithm is proposed in this paper, to overcome the disadvantage of the CIMM algorithm. And the proposed ATPFC-IMM algorithm can adapt to the maneuvering motion of the target.

The proposed ATPFC-IMM algorithm is based on the structure of the classical IMM algorithm. In order to achieve better matching of model probabilities, the model probability information was used in two places on the basis of the classical IMM algorithm: 1. Correction of the probability transfer matrix in real time and 2. Combined with a fuzzy control system to update the model probability again. The processing framework flowchart of the ATPFC-IMM algorithm is shown in Figure 1.

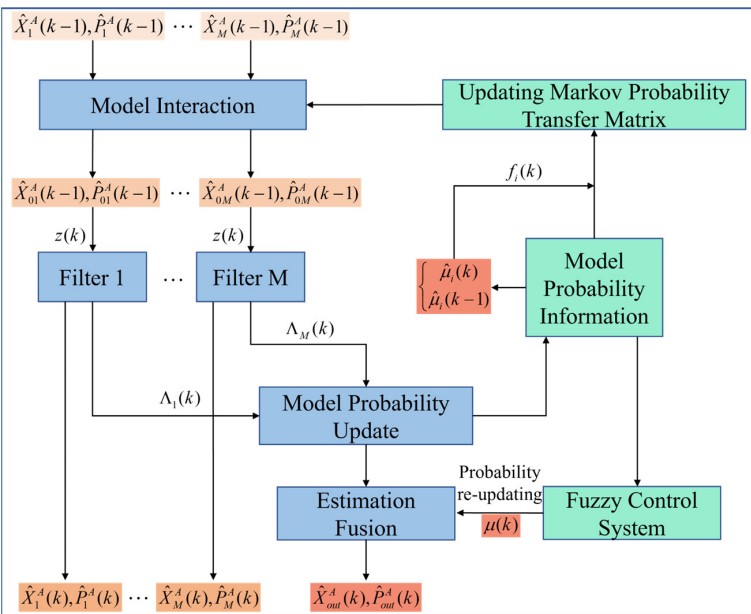

**Figure 1.** The processing framework flowchart of the ATPFC-IMM algorithm.

As shown in Algorithm 1, the processing procedure of the proposed ATPFC-IMM algorithm in this study is primarily based on the classical IMM algorithm. Firstly, after obtaining the preliminary probabilities (by the CIMM algorithm), $\hat{\mu}_i(k), \hat{\mu}_i(k-1)$, and likelihood functions, $\Lambda_k$, of the models, the model with the highest proportion of likelihood functions was determined based on the likelihood function ratio, $Like_{ij}$, of each model. The probability information of this model was then used to construct the correction function, $f_j(k)$. And the probability transfer matrix, $P'_t(k)$, was modified and normalized. In the subsequent time step (radar sampling period), the updated probability transfer matrix, $P_t(k)$, was utilized. Next, the preliminary probabilities, $\hat{\mu}_i(k), \hat{\mu}_i(k-1)$, of the models obtained earlier were inputted into the designed fuzzy control system to reallocate the probabilities for each model. Once the probabilities, $\mu(k)$, of all models were processed, the resulting model probabilities, $\mu_i(k)$, are one of the outputs of the algorithm, which were used for the subsequent state estimation fusion.

---

**Algorithm 1:** Implementation pseudocode for the proposed ATPFC-IMM algorithm in a single cycle.

---

**Background:** The classical IMM algorithm.
**Function:** Accelerating model switching and updating model probabilities.
**1: Input:** The probability information of each model.
**2: Computation:**
    **(1)** Likelihood function ratio:
**for** ($i = 1\ to\ M$) **do**
**for** ($j = 1\ to\ M,\ i \neq j$) **do**
$Like_{ij} = \Lambda_k^i / \Lambda_k^j$;
**end for**
**end for**
    **(2)** Correction function: $f_j(k)$.
**3: Correction of probability transfer matrix:**
**for** ($j = 1\ to\ M$) **do**
$P'_t(k) = f_j(k) \cdot P_t(k)$;
**end for**
**4: Normalization of probability transfer matrix:** $P'_t(k) \rightarrow P_t(k)$.
**5: Re-Updating:**
    Inputting the model probability information into the fuzzy control system:
**for** ($i = 1\ to\ M$) **do**
$\hat{\mu}_i(k), \hat{\mu}_i(k-1) \rightarrow \mu_i(k)$;
**end for**
**6:** The currently obtained model probabilities are kept for the next moment of the algorithm update: $\mu_i(k)$ ($i = 1, 2, \cdots, M$).
**7: Output:** (1). The re-updating probability of each model: $\mu_i(k)$.
(2). Model state estimation fusion: $X_{out}(k) = \sum\limits_{i=1}^{M} \hat{X}_i^A(k) \cdot \mu_i(k)$.
**8: Return** $\mu_i(k), X_{out}(k)$;

---

### 3.1. The Correction of the Probability Transfer Matrix

In the classical IMM algorithm, the probability vector (likelihood function) of the model can reflect the probability of the model, that is, the matching degree of a model and the current motion state of the tracked target [29]. So, the likelihood function can be used to find a model in a model set of the ATPFC-IMM algorithm that best matches the current motion state of the target, and the probability information of this model can be used to construct a correction function, which can be used to adaptively correct the probability transfer matrix in real time [30].

If the ATPFC-IMM algorithm model set includes three models ($M = 3$), the model likelihood function ratio can be expressed by Equation (10).

$$Like_{12} = \Lambda_k^1 / \Lambda_k^2, Like_{13} = \Lambda_k^1 / \Lambda_k^3, Like_{23} = \Lambda_k^2 / \Lambda_k^3 \tag{10}$$

The model that best matches the target-motion state can be obtained by comparing the likelihood functions between the models. By utilizing the three scenarios presented in Equation (11), the values of $a$ and $j$ can be obtained and subsequently employed in Equation (12). The calculation of that equation determines the correction function for the real-time adjustment of the probability transfer matrix.

$$\begin{cases} Like_{12} > 1 \ \& \ Like_{13} > 1 & a = 0, j = 1 \\ \min(Like_{12}, Like_{13}, Like_{23}) = Like_{12} \ a = 1, j = 2 \\ \max(Like_{12}, Like_{13}, Like_{23}) = Like_{12} \ a = 2, j = 3. \end{cases} \tag{11}$$

The determination of the correction function, $f_j(k)$, is defined as follows:

$$\begin{aligned} f_j(k) &= 1/(1 - (\mu_j(k) - \mu_j(k-1)))\cdot \\ &(0.5(2-a)(1-a) + a(2-a) + 0.5a(a-1)) \end{aligned} \tag{12}$$

In some scenarios, adaptivity may lead to the over-modification of the probability transfer matrix, which can lead to anomalies. In order to be able to adjust the adaptivity of the ATPFC-IMM algorithm to the real-time demand, a coefficient, $\lambda$, was designed. The closer the value of this coefficient, $\lambda$, was set to 0, the weaker the adaptability of the ATPFC-IMM algorithm. (In this paper, the value of $\lambda$ was set to 1). Combined with coefficient $\lambda$, the correction function, $\hat{f}_j(k)$, is redefined in Equation (13).

$$\hat{f}_j(k) = f_j(k)^{\lambda} \ (\lambda > 0) \tag{13}$$

The process of using the correction function to correct the probability transfer matrix and normalization is as follows:

$$\begin{aligned} P'_{t_{ij}}(k) &= \hat{f}_j(k)\cdot P_{t_{ij}}(k-1) \ (i, j = 1, 2, \cdots, M) \\ P_{t_{ij}}(k) &= \frac{P'_{t_{ij}}(k)}{\sum_{j=1}^{M} P'_{t_{ij}}(k)} \end{aligned} \tag{14}$$

### 3.2. The Design of the Fuzzy Control System

In a PD radar, it is often necessary to track maneuvering targets. The maneuvering transformation of the target requires the change in the tracking model of the algorithm accordingly, and requires a high update speed [31]. Therefore, in order to meet this requirement, the ATPFC-IMM algorithm introduces a fuzzy control system for reallocating and updating model probabilities.

The fuzzy control system was a mappable system that fully utilized model information in this paper. The current obtained model information was used as the system input, and the updated model probability (system output) was used in the estimation fusion part of the ATPFC-IMM algorithm. The processing of the fuzzy control system is shown in Figure 2, and the design of the fuzzy control system is shown in Figure 3 and Table 1.

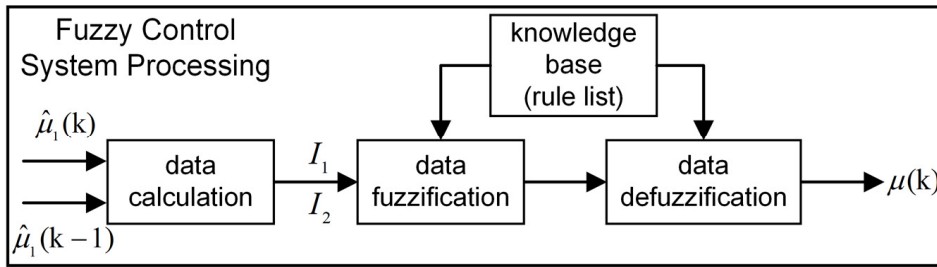

**Figure 2.** The processing of the fuzzy control system.

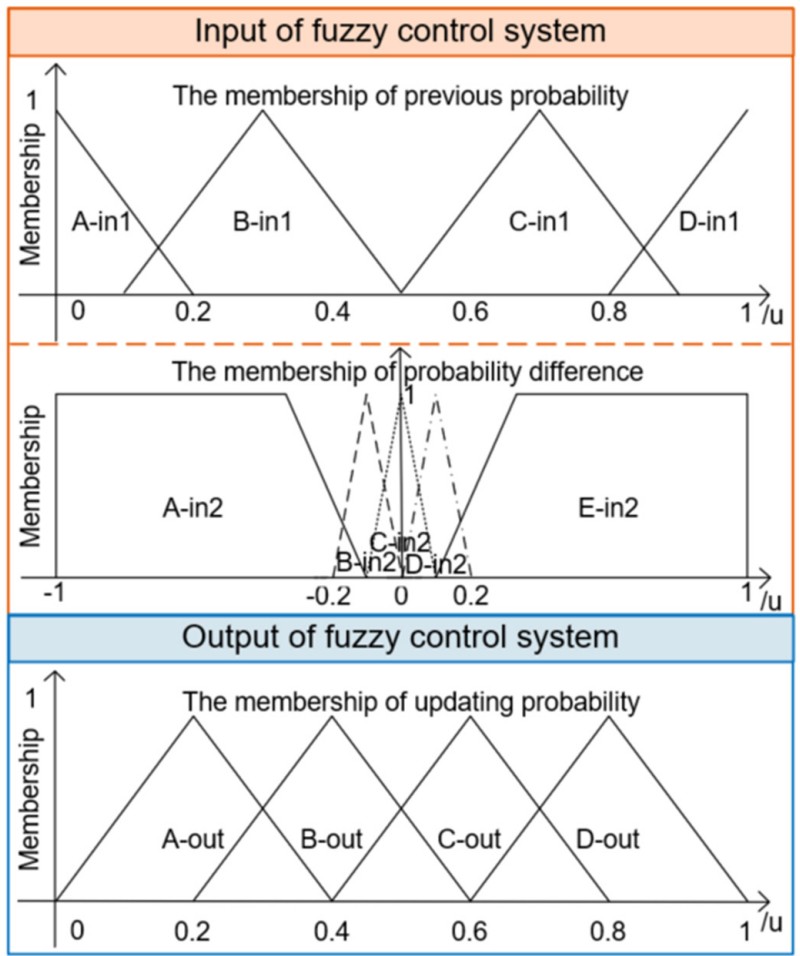

**Figure 3.** The design of the input and output memberships.

**Table 1.** Rule table of the fuzzy control system.

| Output | | Input-2 | | | | |
|---|---|---|---|---|---|---|
| | | **A-in2** | **B-in2** | **C-in2** | **D-in2** | **E-in2** |
| Input -1 | A-in1 | A-out | A-out | A-out | B-out | C-out |
| | B-in1 | A-out | A-out | B-out | C-out | D-out |
| | C-in1 | A-out | B-out | C-out | D-out | D-out |
| | D-in1 | B-out | C-out | D-out | D-out | D-out |

The fuzzy rules of a fuzzy control system are represented as:

$$\text{if Input-1 is } I_i \text{ and Input-2 is } I_j, \text{ THEN Output-1 is } O_i \tag{15}$$

where Input-1 and Input-2 are the input control variables of the fuzzy control system, $I_i$ and $I_j$ are the corresponding fuzzy variables, Output-1 is the output control variable, and $O_i$ is the output fuzzy variable.

The fuzzy control system is an inference synthesis algorithm; it can be expressed as:

$$(A^* \ \& \ B^*)O(A \ \& \ B \rightarrow C) \ = \ (A^* \ \& \ B^*)OR \ = \ C^* \tag{16}$$

$$R \ = \ A^* \ \& \ B^* \rightarrow C^* \tag{17}$$

where $A^*$ and $B^*$ are the fuzzy subsets, $C^*$ is the inference result, $O$ denotes the inference synthesis algorithm, and $R$ denotes the fuzzy rule.

Defuzzification is carried out by using the method of the centroid of the area of the membership function, and its mathematical model can be represented as:

$$u_{cen} = \frac{\int_U Y(u)u\,du}{\int_U Y(u)\,du} \tag{18}$$

where $u_{cen}$ is the value of the horizontal coordinate of the centroid of the area; $Y(u)$ is the membership function of a fuzzy subset in the domain, $U$.

Equations (14)–(18) describe the fuzzification and defuzzification processes of the fuzzy control system, and the inference results of the system can be obtained after performing these two steps. In fact, the processing process of the fuzzy control system can be understood as a special problem combining inference and geometry. It ingeniously simplifies the complex mathematical operations and effectively reduces the computational complexity of probability mapping updates.

In order to further describe the workflow and data processing of the fuzzy control system, an example is presented in this paper. Assume that the current model probability is 0.4, and the previous model probability is 0.3 [30]. According to the membership function of the system input and output in Figure 2, the input of the fuzzy control system can be defined as B-in1 and D-in2; so, the system output is C-out. Taking the minimum membership degree (1 in this example), according to the centroid method, the updated model probability can be obtained as the centroid abscissa of the geometric figure, C-out, which is 0.6. All models in the model set of the ATPFC-IMM algorithm undergo probability mapping updates through the above steps, and the obtained probabilities of each model are used for the estimation fusion part of the ATPFC-IMM algorithm. The basis of the aforementioned fuzzy control system and additional rules are shown in Table 1 (Table 1 contains 20 rules for fuzzy control systems corresponding to the design of the membership function in Figure 3).

The ATPFC-IMM algorithm incorporates a correction function for the real-time calibration of the probability transfer matrix. By updating the model probability again with the fuzzy control system, the model probability of the ATPFC-IMM algorithm can be quickly switched and the model probability of matching target motion can be increased. The two cooperate with each other, and effectively improve the comprehensive performance of the ATPFC-IMM algorithm.

## 4. Simulation and Analysis

In this section, three different sets of simulation experiments are included. In Experiment 1, the effects of the classical IMM algorithm (CIMM), ATPM-PIMM [3] algorithm, and the proposed ATPFC-IMM algorithm on the model switching speed and the tracking error are analyzed. In Experiment 2, in order to explore the possible effects of different maneuver modes or maneuver intensities of the simulated target on the CIMM algorithm, ATPM-PIMM [3] algorithm, and the proposed ATPFC-IMM algorithm, the maneuvering performance of the target is further improved. In Experiment 3, a set of complex maneuver data in the PD radar are selected. Moreover, the CIMM, ATPM-PIMM [3], and the proposed ATPFC-IMM algorithms are used to track maneuver targets in the PD radar to demonstrate the effectiveness of the ATPFC-IMM algorithm. In the three experiments, a real measurement process with the addition of random process noise is simulated to analyze the robustness of the ATPFC-IMM algorithm. It has to be noted that, since the threshold for algorithm switching is involved in the ATPM-PIMM [3] algorithm, in this paper, the threshold value (0.8) with a good comprehensive performance was used during the simulation.

Additionally, the root mean square error (RMSE) was utilized to evaluate the effectiveness of the algorithms [32]. It is defined as follows [33]:

$$RMSE'(n) = \left\{ \frac{1}{M} \sum_{j=1}^{M} \|x_j(n) - \hat{x}_j(n)\|_2^2 \right\}^{1/2} \tag{19}$$

$$RMSE = \frac{1}{N} \sum_{n=1}^{N} RMSE'(n) \tag{20}$$

where $x_j(n)$ denotes the true state of the target at moment $j$, $n$ denotes the number of Monte Carlo simulations, $\hat{x}_j(n)$ denotes the target state estimated by filtering the target at moment $j$, $M$ denotes the total number of sampling cycles during the simulation, and $N$ indicates the number of iterations of the Monte Carlo simulation.

### 4.1. Experiment 1

Experiment 1 was set as follows: two-model set: (1) constant-velocity model (CV) and (2) coordinated turning model (CT). The target performed coordinated turning motions at 1–100 s and 201–300 s, and constant-velocity motions at 101–200 s and 301–400 s. The initial probability of the CV model and CT model was 0.5; in addition, the CT model had a constant angular rate ($\omega$ = 0.035 rad/s). The sampling time was $T$ = 1 s, the total sampling time was $M$ = 400 s, and the iteration of the Monte Carlo simulation was 100. The probability transfer matrix, $P_t$, of the CIMM, ATPM-PIMM [3], and ATPFC-IMM algorithms is set as:

$$P_t = \begin{bmatrix} 0.9 & 0.1 \\ 0.1 & 0.9 \end{bmatrix} \tag{21}$$

The observation matrix, $H$, of the system is set as:

$$H = \begin{bmatrix} 1 & 0 & 0 & 0 \\ 0 & 0 & 1 & 0 \end{bmatrix} \tag{22}$$

The state transfer matrix, $F_{CV}$, and prediction-process noise matrix, $G_{CV}$, of the CV model are:

$$F_{CV} = \begin{bmatrix} 1 & T & 0 & 0 \\ 0 & 1 & 0 & 0 \\ 0 & 0 & 1 & T \\ 0 & 0 & 0 & 1 \end{bmatrix}; \; G_{CV} = \begin{bmatrix} T^2/2 & 0 \\ T & 0 \\ 0 & T^2/2 \\ 0 & T \end{bmatrix}. \tag{23}$$

The state transfer matrix, $F_{CT}$, and prediction-process noise matrix, $G_{CT}$, of the CT model are:

$$F_{CT} = \begin{bmatrix} 1 & \sin(\omega T)/\omega & 0 & (\cos(\omega T)-1)/\omega \\ 0 & \cos(\omega T) & 0 & -\sin(\omega T) \\ 0 & (1-\cos(\omega T))/\omega & 1 & \sin(\omega T)/\omega \\ 0 & \sin(\omega T) & 0 & \cos(\omega T) \end{bmatrix}; \; G_{CT} = \begin{bmatrix} T^2/2 & 0 \\ T & 0 \\ 0 & T^2/2 \\ 0 & T \end{bmatrix}. \tag{24}$$

The influence of various noises on the radar system in the real environment was simulated, which led to the error of the measurement of the radar system. The measurement noise was added to the measurement model of the simulation process, as shown in the following equation:

$$Z(k) = H(k)X(k) + V(k) \tag{25}$$

$$V(k) = \begin{bmatrix} 50 & 0 \\ 0 & 50 \end{bmatrix} \begin{bmatrix} randn & 0 \\ 0 & randn \end{bmatrix} \tag{26}$$

where *randn* denotes a random scalar obtained from a standard normal distribution.

In Figure 4, the switching and probability error distribution of each model probability when the target-motion model is changed can be seen. The moments when the target-

motion-state model switches are 100 s, 200 s, and 300 s, the fastest model switching is achieved by the ATPFC-IMM algorithm, followed by the ATPM-PIMM [3] algorithm, then the CIMM algorithm. And the most optimal distribution of the corresponding model probability is achieved by the ATPFC-IMM algorithm, followed by the ATPM-PIMM [3] algorithm, then the CIMM algorithm.

As shown in Table 2 and Figure 4, the ATPFC-IMM algorithm has the fastest model switching speed, so the response time of the models is reduced by the ATPFC-IMM algorithm. In addition, both the ATPM-PIMM [3] algorithm and the ATPFC-IMM algorithm can improve the probability of the corresponding matching model relative to the CIMM algorithm, but the ATPFC-IMM algorithm has a better enhancement effect than the ATPM-PIMM [3] algorithm, making the probability of the corresponding model closer to 1. Therefore, the tracking performance in this experiment is effectively improved by the proposed ATPFC-IMM algorithm; the tracking error is reduced compared with the CIMM algorithm and ATPM-PIMM [3] algorithm.

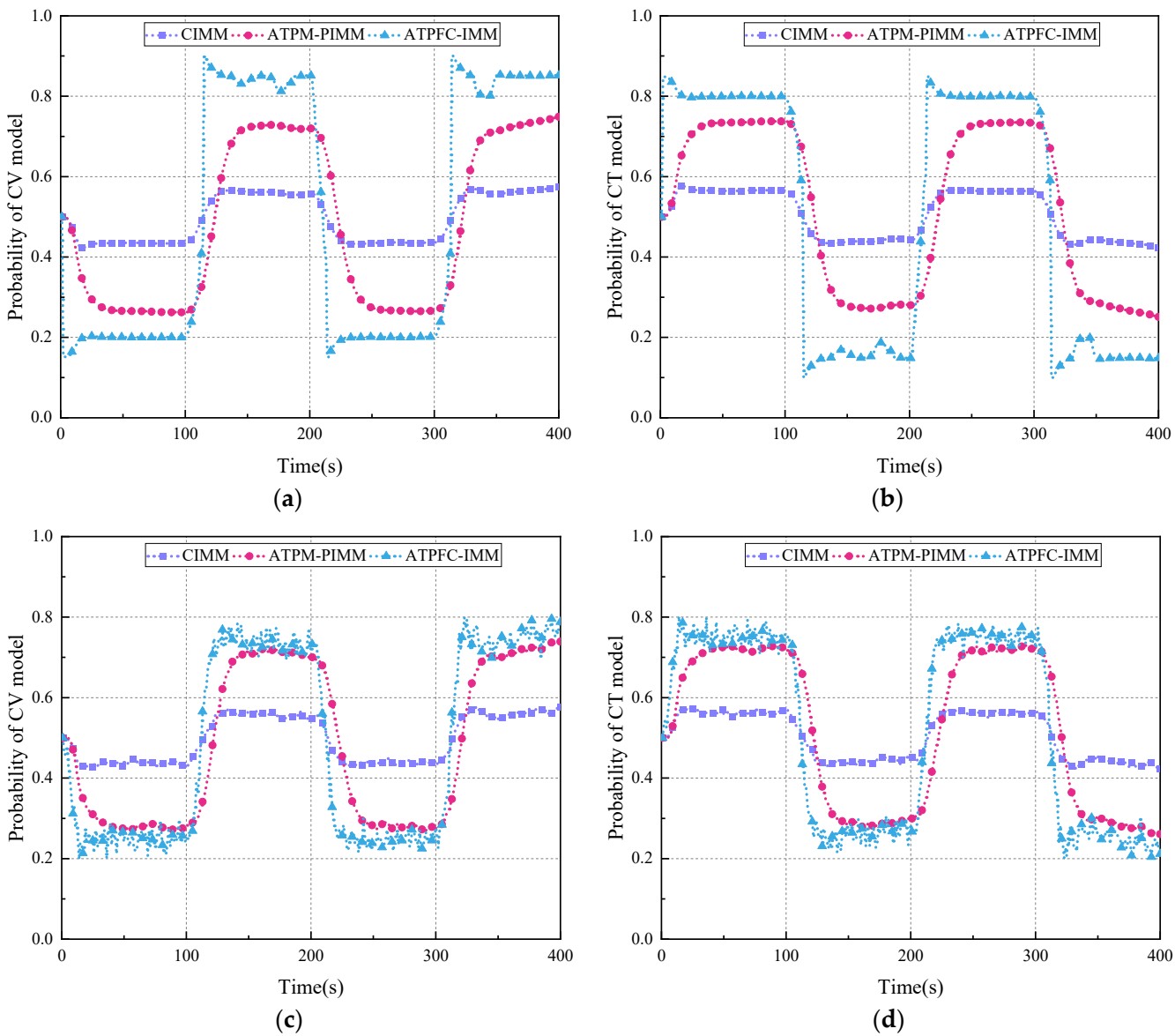

**Figure 4.** *Cont.*

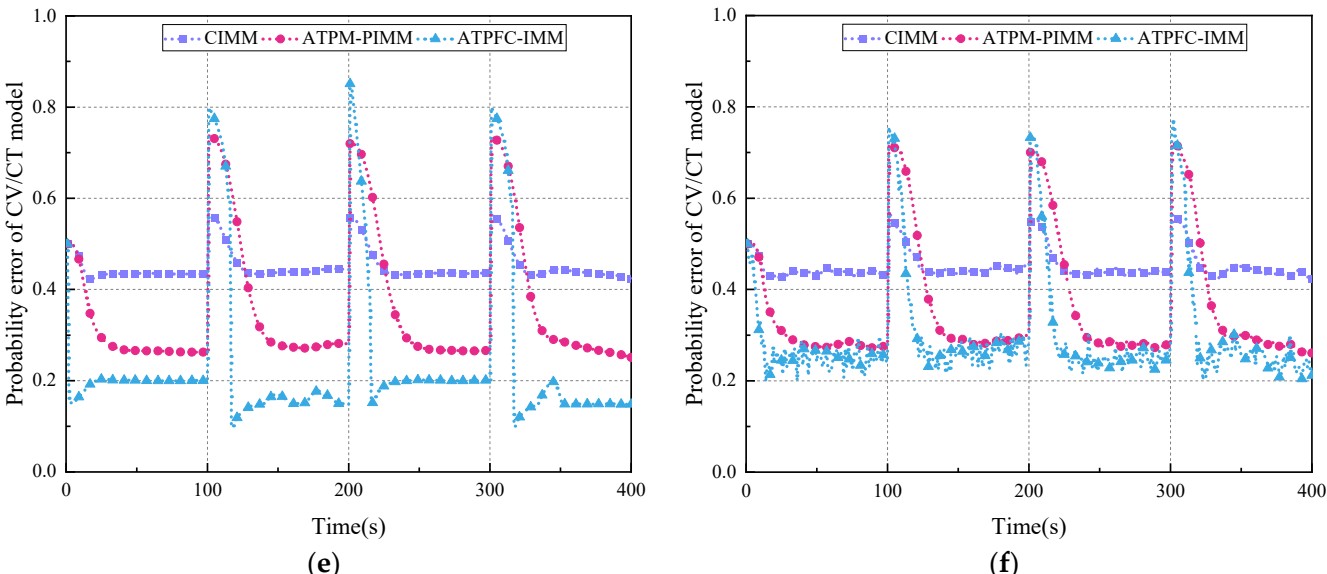

(**e**)                                                    (**f**)

**Figure 4.** Model probability and probability error distribution for Experiment 1. (**a**) Probability of CV model without measurement noise; (**b**) probability of CT model without measurement noise; (**c**) probability of CV model with measurement noise; (**d**) probability of CT model with measurement noise; (**e**) probability error of models without measurement noise; (**f**) probability error of models with measurement noise.

**Table 2.** RMSEs of position tracking in Experiment 1.

| Algorithm | Measurement Noise | RMSE of x-Axis/(m) | RMSE of y-Axis/(m) | RMSE of Position/(m) |
|---|---|---|---|---|
| CIMM | Unadded | 7.6171 | 11.9254 | 15.3929 |
| ATPM-PIMM [3] | | 6.6795 | 10.6392 | 13.7303 |
| ATPFC-IMM | | 6.4601 | 10.4568 | 13.3856 |
| CIMM | Added | 16.5781 | 19.5356 | 26.2282 |
| ATPM-PIMM [3] | | 15.9144 | 18.6876 | 25.2220 |
| ATPFC-IMM | | 15.7616 | 18.5622 | 24.8715 |

The probability transfer matrix, $P_t$, is a very important factor in all types of IMM algorithms, and the setting of $P_t$ often depends on a priori experience. Different $P_t$ values may affect the tracking performance of IMM tracking algorithms. In order to investigate the effect of this factor on such IMM algorithms, a control experiment is set up and $P_t$ is designed in Equation (27). The other experimental parameters were the same as in the previous experiment. The model probability switching for the control experiment is shown in Figure 5.

$$P_t = \begin{bmatrix} 0.99 & 0.01 \\ 0.01 & 0.99 \end{bmatrix} \qquad (27)$$

As shown in Figure 5a–d, the new probability transfer matrix improves the probability when the model matches the target's motion state. The probability of matching the model is closer to 1 compared to the previous experiment, even after the addition of the measurement noise. In addition, as can be seen in Figure 5e,f, the probability error of the models are at a lower level. Overall, the ATPFC-IMM algorithm still outperforms the ATPM-PIMM [3] and CIMM algorithms.

The position error of the tracking process of the corresponding target in Figure 5 is recorded in Table 3. Compared with Table 2, the tracking error of each algorithm decreases

by a certain level, which is directly related to the improvement of the probability of the matching model in Figure 5.

**Table 3.** RMSEs of position tracking in control trial.

| Algorithm | Measurement Noise | RMSE of x-Axis/(m) | RMSE of y-Axis/(m) | RMSE of Position/(m) |
|-----------|-------------------|--------------------|--------------------|----------------------|
| CIMM | Unadded | 5.3504 | 5.6552 | 8.6531 |
| ATPM-PIMM [3] | | 4.7714 | 5.4412 | 8.0322 |
| ATPFC-IMM | | 3.0197 | 3.1945 | 4.9433 |
| CIMM | Added | 13.7502 | 14.2969 | 20.1896 |
| ATPM-PIMM [3] | | 13.4140 | 14.0470 | 19.8215 |
| ATPFC-IMM | | 9.0937 | 9.5531 | 13.3783 |

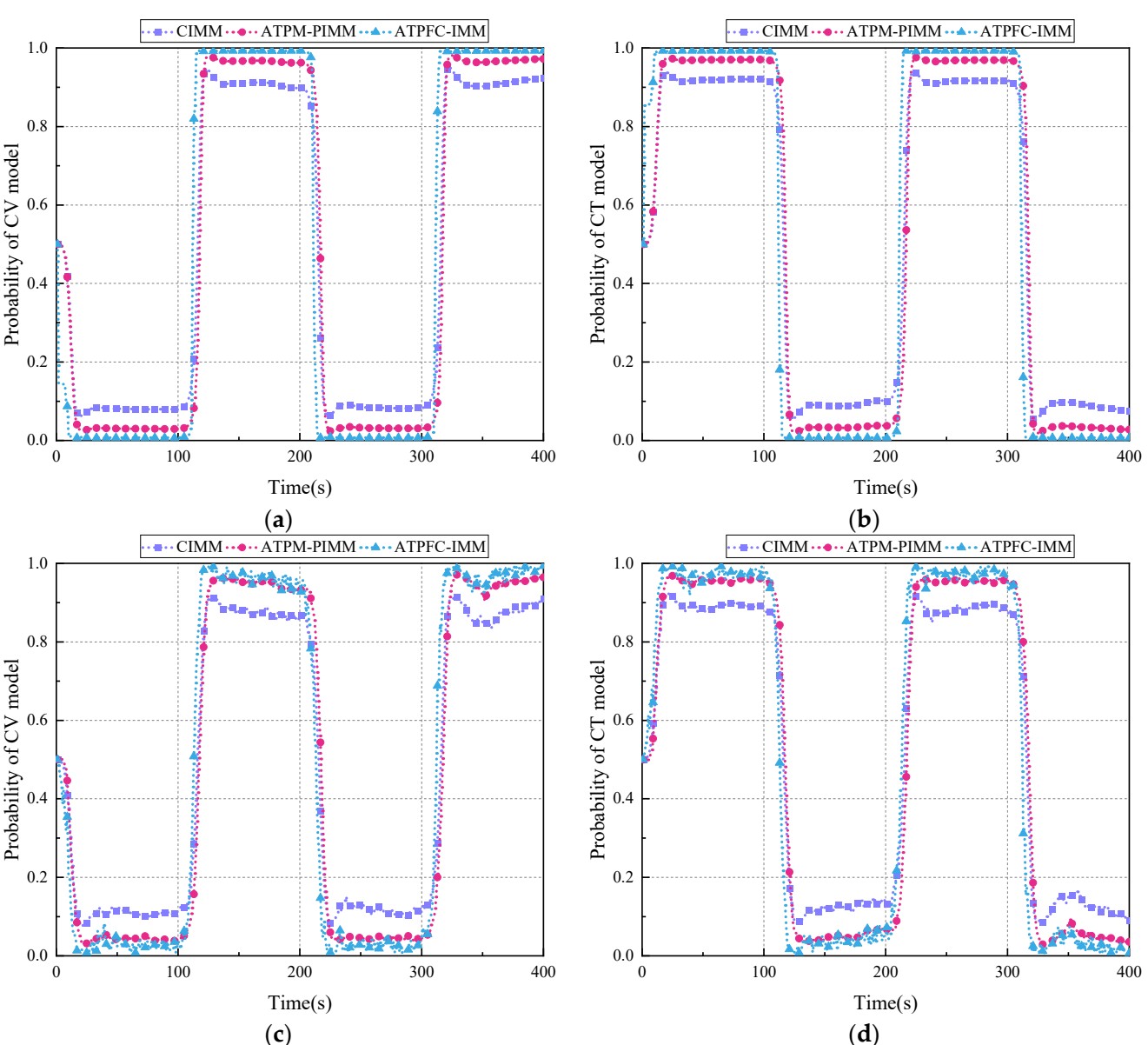

**Figure 5.** *Cont.*

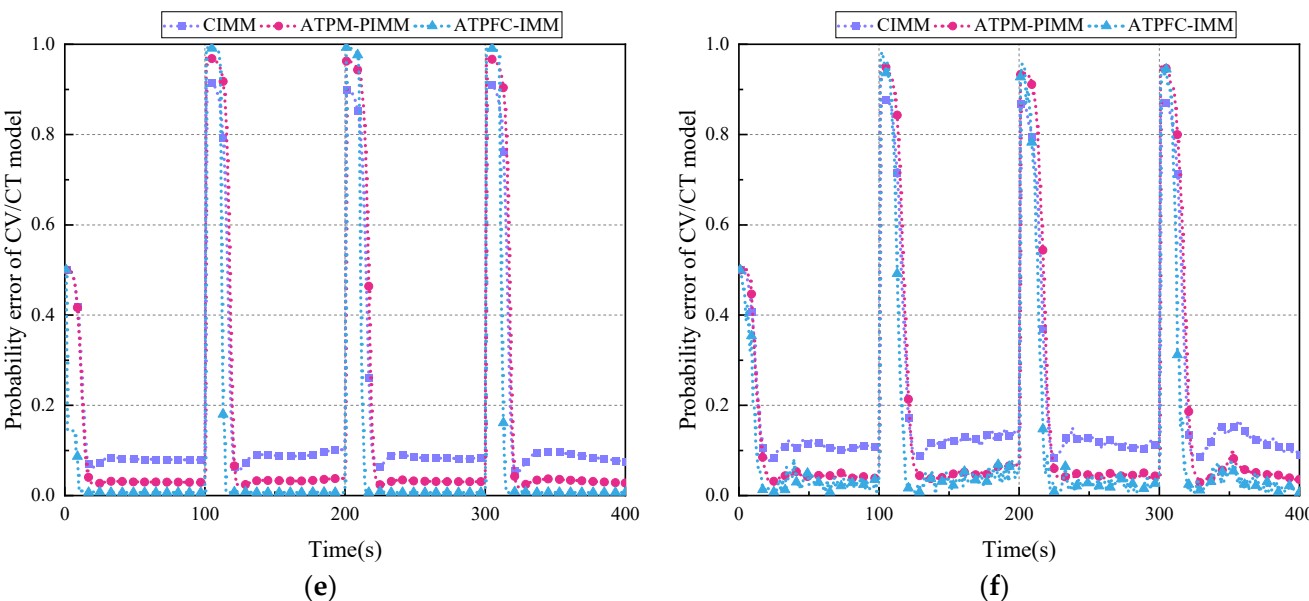

**Figure 5.** Distribution of model probability and probability errors in control trial. (**a**) Probability of CV model without measurement noise; (**b**) probability of CT model without measurement noise; (**c**) probability of CV model with measurement noise; (**d**) probability of CT model with measurement noise; (**e**) probability error of models without measurement noise; (**f**) probability error of models with measurement noise.

### 4.2. Experiment 2

In this subsection, the target had a more complex maneuvering pattern than in Experiment 1, first making a turning motion to the right that lasted 50 s, then making a turning motion to the left that also lasted 50 s, making the two motions four times in a row for a total of 400 s. The motion trajectory of the target is shown in Figure 6.

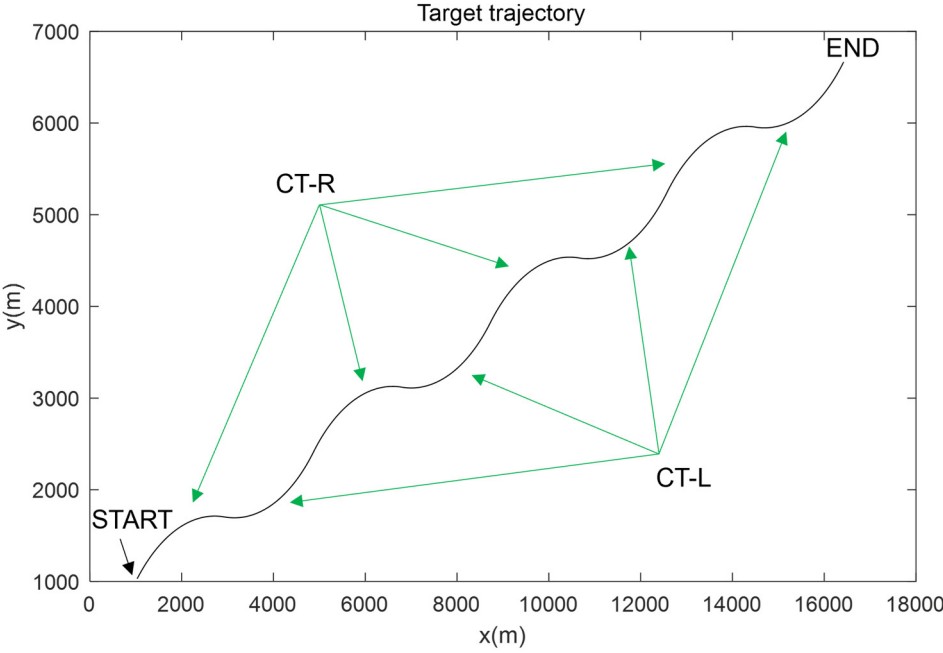

**Figure 6.** Maneuvering target-motion trajectory with two turning motions.

The model set was designed as follows: CT model with a constant angular rate ($\omega = 0.035$ rad/s; this parameter controlled turning to the right: CT-R) and CT model with

a constant angular rate ($\omega$ = −0.035 rad/s; this parameter controlled turning to the left: CT-L). The configuration parameters of the algorithms were the same as in Experiment 1.

First, the probability transfer matrix was designed according to Equation (21) and experiments without adding measurement noise versus adding measurement noise were designed and recorded as Experiment 2-1; then, the probability transfer matrix was designed according to Equation (27) and the same experiments were conducted and recorded as Experiment 2-2.

In Figure 7, the model probability distribution and model probability error of the tracking algorithms can be seen for each tracking algorithm when the maneuvering target-motion state is switched. Figure 7 shows that the model probability in the ATPFC-IMM algorithm better matches the real situation of the target, and the model probability error is minimized accordingly. The three algorithms with good performance are the ATPFC-IMM algorithm, ATPM-PIMM [3] algorithm, and CIMM algorithm in that order. In Table 4, the ATPM-PIMM [3] algorithm is able to slightly reduce the tracking error compared to the CIMM algorithm, and the tracking error of ATPFC-IMM is further reduced.

For the sake of brevity of the experimental part, only the tracking error of Experiment 2-2 was counted in this paper. In Table 5, the tracking performances of the three algorithms in this experiment perform the same as in Table 4. The ATPFC-IMM algorithm has the lowest tracking error, followed by the ATPM-PIMM [3] algorithm and the CIMM algorithm.

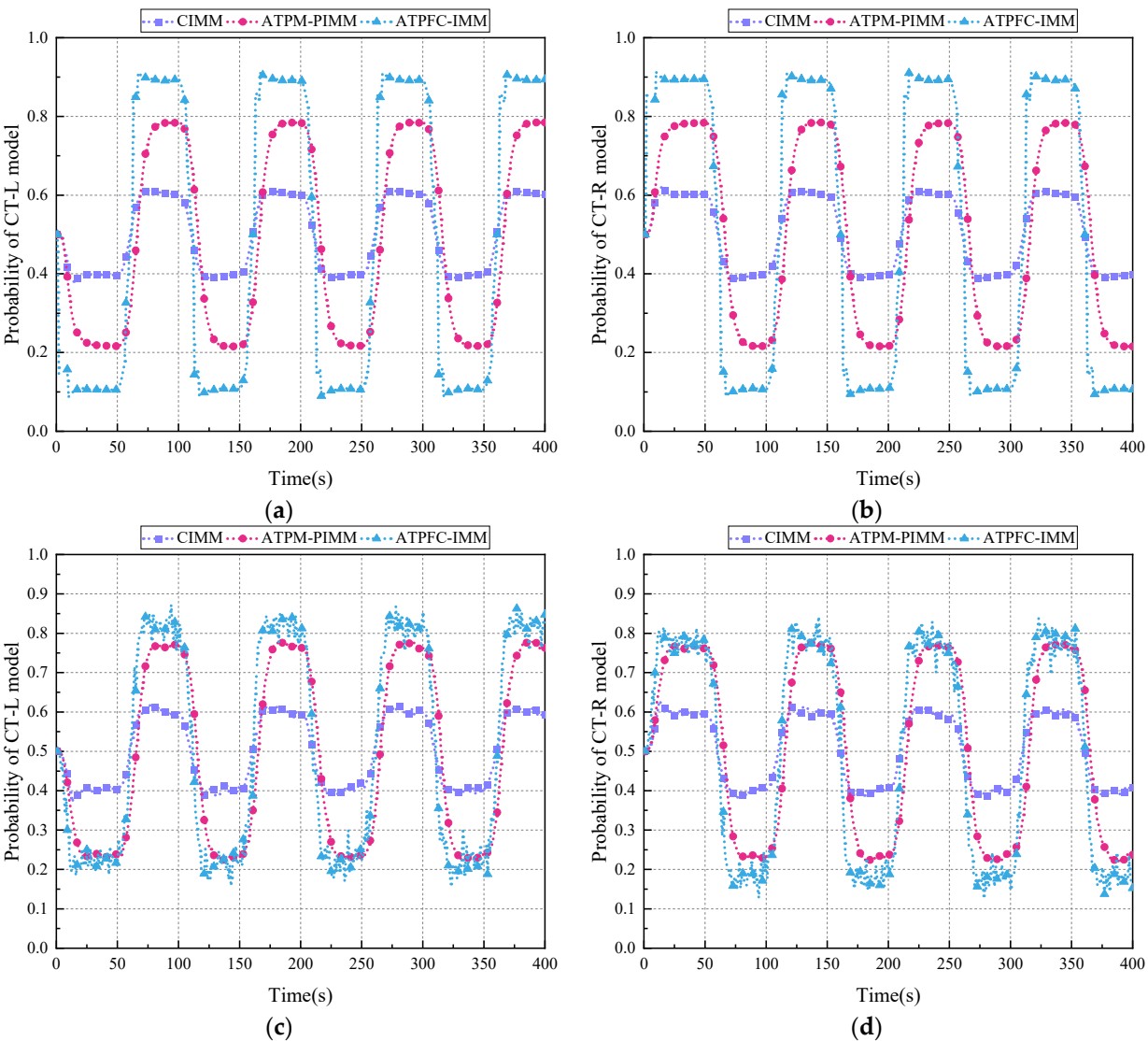

**Figure 7.** *Cont.*

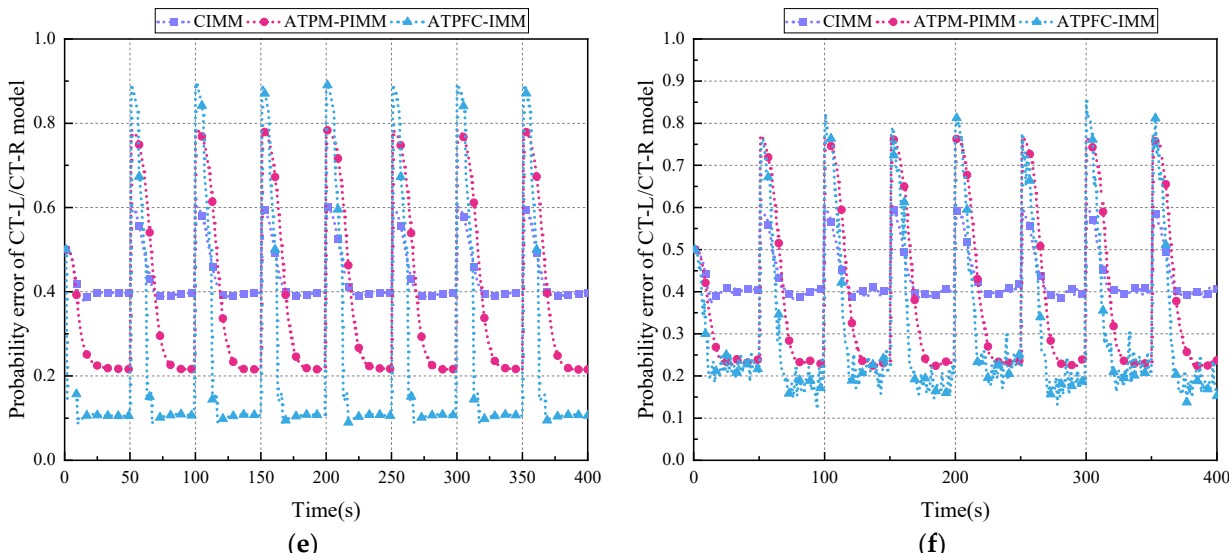

**Figure 7.** Distribution of model probability and probability errors in Experiment 2. (**a**) Probability of CT-L model without measurement noise; (**b**) probability of CT-R model without measurement noise; (**c**) probability of CT-L model with measurement noise; (**d**) probability of CT-R model with measurement noise; (**e**) probability error of models without measurement noise; (**f**) probability error of models with measurement noise.

**Table 4.** RMSEs of position tracking in Experiment 2-1.

| Algorithm | Measurement Noise | RMSE of x-Axis/(m) | RMSE of y-Axis/(m) | RMSE of Position/(m) |
|---|---|---|---|---|
| CIMM | | 6.3057 | 12.7035 | 14.9693 |
| ATPM-PIMM [3] | Unadded | 6.1202 | 12.8428 | 14.8355 |
| ATPFC-IMM | | 4.6364 | 9.5192 | 11.0814 |
| CIMM | | 15.7992 | 23.9586 | 28.8754 |
| ATPM-PIMM [3] | Added | 15.8645 | 23.8320 | 28.8369 |
| ATPFC-IMM | | 13.9693 | 21.0919 | 25.4698 |

**Table 5.** RMSEs of position tracking in Experiment 2-2.

| Algorithm | Measurement Noise | RMSE of x-Axis/(m) | RMSE of y-Axis/(m) | RMSE of Position/(m) |
|---|---|---|---|---|
| CIMM | | 4.7026 | 7.1481 | 9.2974 |
| ATPM-PIMM [3] | Unadded | 4.4345 | 6.5464 | 8.5476 |
| ATPFC-IMM | | 3.6148 | 4.7559 | 6.5612 |
| CIMM | | 12.8534 | 19.4745 | 23.6441 |
| ATPM-PIMM [3] | Added | 11.9888 | 18.4349 | 22.3480 |
| ATPFC-IMM | | 11.6783 | 17.7896 | 21.5809 |

### 4.3. Experiment 3

In this scenario, a strong maneuvering target in real applications was simulated. A set of complex maneuvering unmanned aerial vehicle (UAV) target trajectory data observed by pulse-Doppler radar was used as the real target trajectory. The ATPFC-IMM algorithm, ATPM-PIMM [3] algorithm, and CIMM algorithm were used to observe this UAV target.

In this experiment, a CT model ($\omega$ = −0.35 rad/s) was added; at this time, the model set included a CV model, two CT models ($\omega$ = ±0.35 rad/s), and the sampling period was 1 s; the transfer equation of the UAV target state and the measurement noise were the same as in Experiment 1. This experiment mainly analyzed the tracking of complex maneuvering targets by each target tracking algorithm, and the probability transfer matrix, $P_t$, is designed as follows:

$$P_t = \begin{bmatrix} 0.8 & 0.1 & 0.1 \\ 0.1 & 0.8 & 0.1 \\ 0.1 & 0.1 & 0.8 \end{bmatrix} \tag{28}$$

Based on the above simulation experimental conditions, the tracking results of the ATPFC-IMM algorithm, ATPM-PIMM algorithm, and CIMM algorithm are shown in Figure 8. In order to investigate the robustness of the ATPFC-IMM algorithm proposed in this paper in tracking complex maneuvering targets, the same measurement noise as in Experiment 1 was added to the measurement process tracked by each target tracking algorithm. In the two zoomed-in areas marked, it can be seen that the tracked trajectories of the ATPM-PIMM and CIMM algorithms slightly deviate from the true trajectory of the target. In addition, the tracked trajectory of the ATPFC-IMM algorithm is closer to the true trajectory of the maneuvering target with the UAV. The RMSEs of the tracked trajectories of the above three algorithms are shown in Table 6.

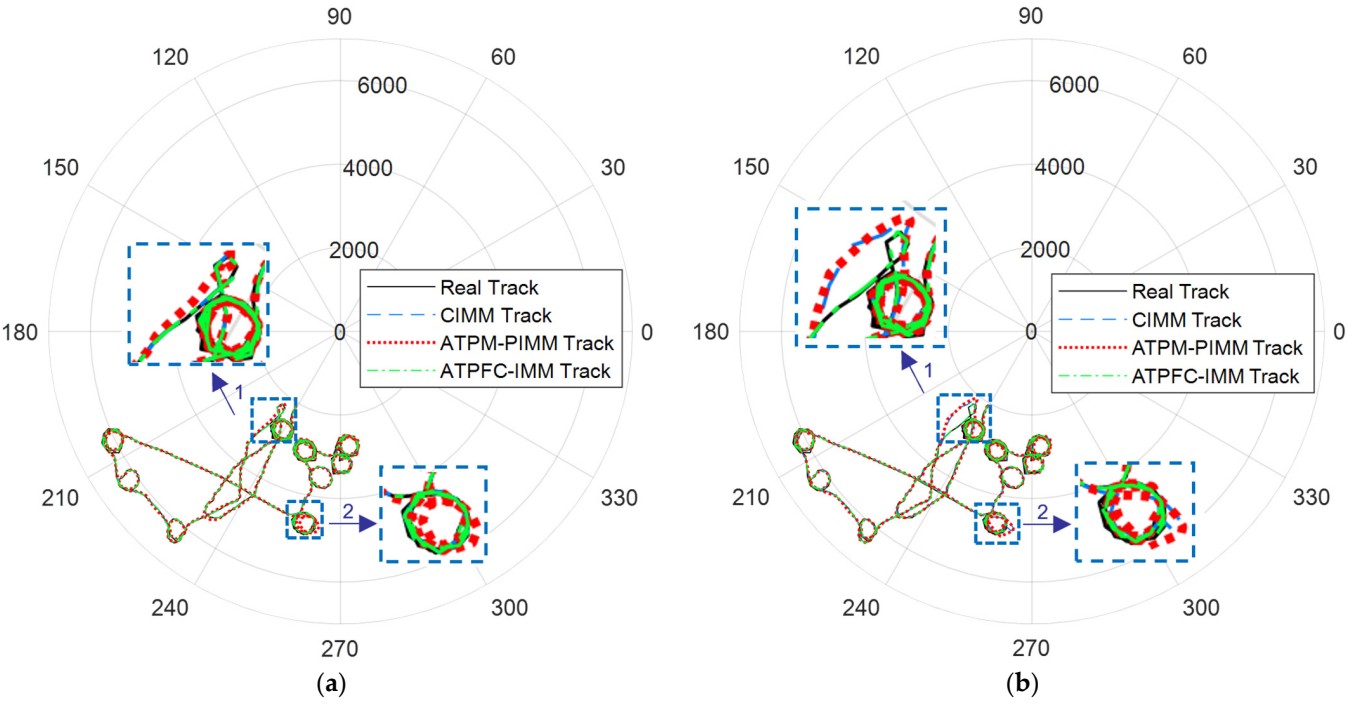

**Figure 8.** The tracking trajectory of three algorithms for the maneuvering target. (**a**) The tracking trajectory of three algorithms without measurement noise. (**b**) The tracking trajectory of three algorithms with measurement noise.

In Figure 8b, it can be seen that both the ATPM-PIMM [3] algorithm and the CIMM algorithm are affected by the noise of the measurement process, the tracked trajectories of the ATPM-PIMM [3] algorithm and the CIMM algorithm in Figure 8b deviate from the target's real trajectory to an increased extent, and the ATPFC-IMM algorithm exhibits a better tracking performance. The RMSEs of the tracked trajectories for each algorithm with measurement noise are also recorded in Table 6.

In Table 6, the tracking errors of all three algorithms increase after adding measurement noise; the position tracking accuracy is improved by 5.7% in the ATPM-PIMM [3] algorithm

compared with the CIMM algorithm, and improved by 19.6% in the ATPFC-IMM algorithm compared with the CIMM algorithm.

**Table 6.** RMSEs of position tracking in Experiment 3.

| Algorithm | Measurement Noise | RMSE of x-Axis/(m) | RMSE of y-Axis/(m) | RMSE of Position/(m) |
|---|---|---|---|---|
| CIMM | Unadded | 33.5938 | 37.0065 | 55.7760 |
| ATPM-PIMM [3] | | 28.8964 | 33.7882 | 49.5958 |
| ATPFC-IMM | | 20.9961 | 22.8124 | 33.8124 |
| CIMM | Added | 58.1596 | 56.8049 | 82.7898 |
| ATPM-PIMM [3] | | 54.6615 | 54.1209 | 78.0547 |
| ATPFC-IMM | | 46.5514 | 46.9047 | 66.5357 |

## 5. Application and Analysis

This section mainly focuses on the application effectiveness of the ATPFC-IMM algorithm. The CIMM algorithm and the ATPFC-IMM algorithm were applied to the data processing of the actual PD radar, containing target-motion scenarios. The targets performed circular maneuvers and continuous-turning maneuvers in the scenarios. The track information on the radar terminal can reflect the tracking performance of both algorithms for complex maneuvering targets in the real environment.

The PD radar utilized in this study worked at the frequency of 9.6–12 GHz. A transmitting power of 320 W, a minimum detectable range of 150 m, a range resolution of 30 m, both azimuth and elevation resolutions of 0.6°, a data rate of 6 s, a maximum detection distance of 20 km, and a beam coverage of 360° were achieved. Furthermore, this PD radar was capable of detecting and tracking target velocities in the range of 0.3 m/s to 100 m/s. The physical diagram of the radar is shown in Figure 9, while more detailed parameters are presented in Table 7.

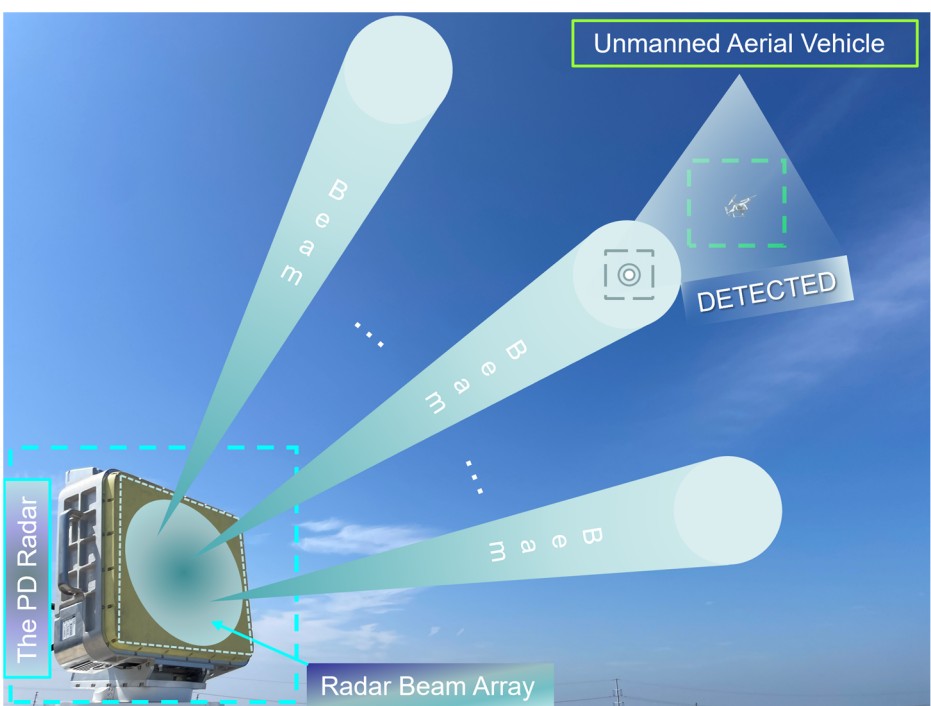

**Figure 9.** PD radar for UAV observation application scenarios.

**Table 7.** Some important indicators and parameters of the PD radar in this application.

| Indicators | Specifications | Indicators | Specifications |
|---|---|---|---|
| Range | >7 km | Height | >400 m |
| Ranging Accuracy | <15 m | Azimuth Accuracy | <0.6° |
| Pitch Accuracy | <0.6° | Distance Resolution | <30 m |

A model set of six models was configured, including two CV models, two CT models, a CS model, and a CA model [14]. The probability transfer matrix, $P_t$, is set as:

$$P_t = \begin{bmatrix} 0.5 & 0.1 & 0.1 & 0.1 & 0.1 & 0.1 \\ 0.1 & 0.5 & 0.1 & 0.1 & 0.1 & 0.1 \\ 0.1 & 0.1 & 0.5 & 0.1 & 0.1 & 0.1 \\ 0.1 & 0.1 & 0.1 & 0.5 & 0.1 & 0.1 \\ 0.1 & 0.1 & 0.1 & 0.1 & 0.5 & 0.1 \\ 0.1 & 0.1 & 0.1 & 0.1 & 0.1 & 0.5 \end{bmatrix} \tag{29}$$

In this section, the maneuverable target of interest was replaced with the DJI Phantom 4 drone, it is displayed in the Figure 10. This drone has a maximum flight speed of 20 m/s, ascent speed of up to 6 m/s, descent speed of up to 3 m/s, and a maximum flight altitude of 6000 m. The drone was launched in an air space approximately 5 km away from the radar. The trajectory information of the UAV target observed by the PD radar was displayed on the radar terminal in real time.

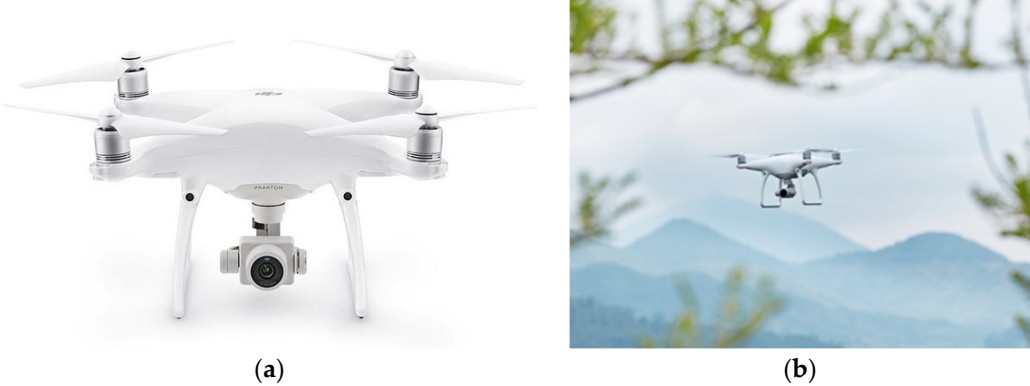

(**a**)          (**b**)

**Figure 10.** The UAV target in PD radar observation scenarios. (**a**) DJI Phantom 4 drone; (**b**) an example of UAV flight form.

*5.1. Case 1 (Circular Maneuvering)*

In this case, the observation group included a target with complex maneuvers, primarily performing a series of irregular curved movements, circular maneuvers, and related maneuvers. The unmanned aerial vehicle (UAV) performed maneuvers in the airspace approximately 5 km away from the radar installation site. Considering this specific UAV as the target of observation, its initial flight speed at this position was estimated to be around 20 m/s. During maneuvering, the speed may decrease, and the direction vector of the velocity can also undergo changes. The direction of velocity is defined as positive when moving away from the radar and negative when moving towards the radar. The tracking results are shown in Figure 11.

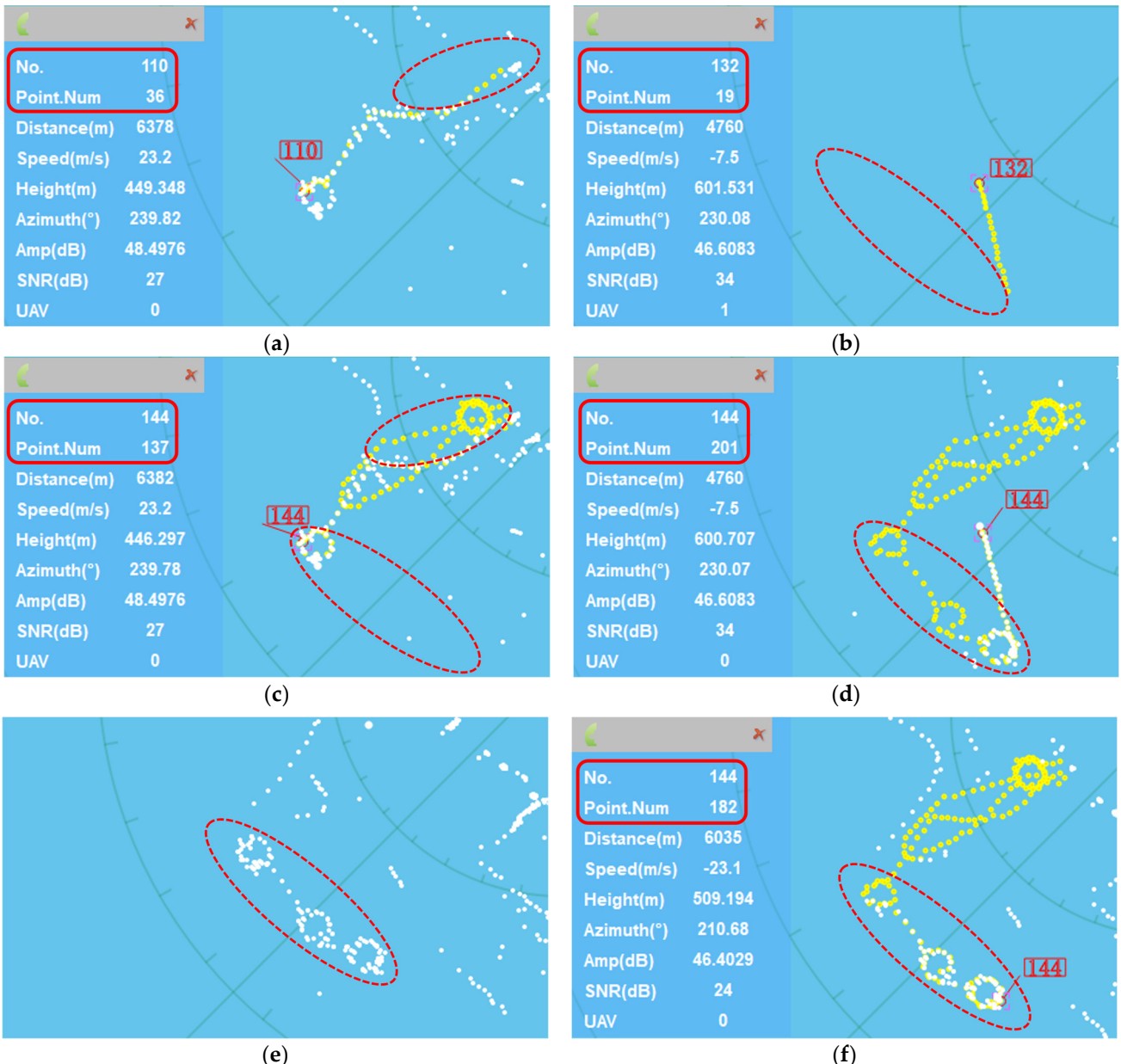

**Figure 11.** Radar terminal displays of CIMM algorithm and ATPFC-IMM algorithm during circular maneuvering of the target. (**a**) The first-half process in the CIMM algorithm; (**b**) the second-half process in the CIMM algorithm; (**c**) the first-half process in the ATPFC-IMM algorithm; (**d**) the second-half process in the ATPFC-IMM algorithm; (**e**) a circular maneuvering area of the CIMM algorithm; (**f**) a circular maneuvering area of the ATPFC-IMM algorithm.

In Figure 11, the target trajectory information is presented, including the trajectory number and the number of target points in the trajectory (trajectory length). This study primarily focused on the number of target points in the trajectory (Point.Num). In Figure 11a, the target performs a series of circular maneuvers, but no trajectory information is available (target tracking lost). The target continues the circular maneuver, with a trajectory number of 110 and 36 target points in the trajectory. In Figure 11b, it can be observed that the trajectory number is no longer 110 but 132, with 19 target points in the trajectory, indicating a trajectory interruption (target tracking lost). Additionally, no trajectory is generated in the red elliptic region (target circular maneuver), as shown in Figure 11e, which indicates severe target tracking loss in the CIMM algorithm in this case.

The application of the ATPFC-IMM algorithm is demonstrated in Figure 11c,d,f. In Figure 11c, the target track number is 144 and the trajectory length is 137. After the maneuver of the target, the observed trajectory is displayed in Figure 11d, with the track index still being 144, and the track length (Point.Num) increasing to 201. (The white points represent the original radar detections, while yellow points represent the target points in the trajectory). Continuous tracking is also maintained in the maneuvering area in Figure 11f. Therefore, compared to Figure 11a,b,e (the application of the CIMM algorithm), at the same time instance, this trajectory has significantly more target points, enabling the continuous tracking of the maneuvering target.

### 5.2. Case 2 (Continuous-Turning Maneuvers)

In case 2, the CIMM algorithm and ATPFC-IMM algorithm were applied to the PD radar after observing a group of targets performing continuous-turning maneuvers. The radar terminal displays corresponding to the CIMM algorithm and ATPFC-IMM algorithm are shown in Figures 12 and 13, respectively. Similarly, the comparative analysis of relevant tracking information on the radar terminal enables the assessment of algorithmic performance.

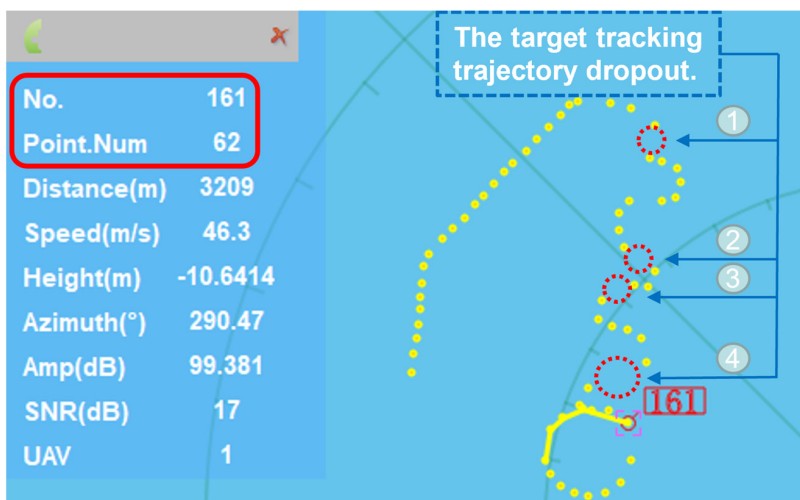

**Figure 12.** The display of the CIMM algorithm on the radar terminal during continuous-turning maneuvers of the target (throughout the entire maneuver cycle).

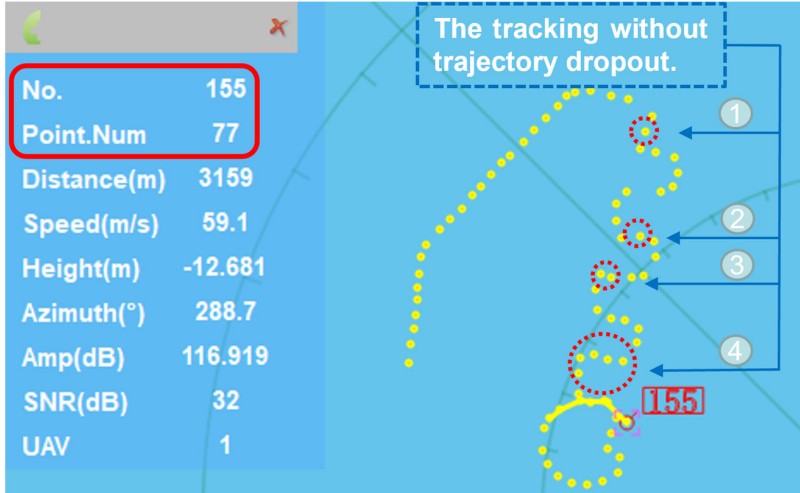

**Figure 13.** The display of the ATPFC-IMM algorithm on the radar terminal during continuous-turning maneuvers of the target (throughout the entire maneuver cycle).

The application of the CIMM algorithm in case 2 is illustrated in Figure 12, where the track number of the maneuvering target is identified as 161, encompassing 62 target points. Additionally, it is observed that the tracking experiences point dropouts at locations with four track losses in the vicinity of target maneuvers. The intermittent loss of target tracking results in a certain level of sparsity in the overall track representation on the terminal and leads to the failure of fully capturing the continuous-turning maneuvering target.

The application of the ATPFC-IMM algorithm in this particular case is shown in Figure 13. To facilitate a more rigorous comparison between the tracking performance of the ATPFC-IMM and CIMM algorithms for maneuvering targets, a snapshot of the radar display terminal in Figure 13 is taken at the same moment as Figure 12. No target points were detected at the marked positions in Figure 12, particularly at the fourth marked position where five target points were lost. When setting the radar's data working rate at 6 s, the CIMM algorithm effectively resulted in a target loss phenomenon lasting for 20 s to 30 s at the fourth marked position. Conversely, Figure 13 reveals the presence of points in the regions marked throughout the target maneuvering area. In Figure 13, the track number of the target is identified as 155, comprising 77 target points. Figure 12 (corresponding to the CIMM algorithm) only includes 62 target points; it indicates a 24.2% improvement in the track length (continuity of radar tracking) by the ATPFC-IMM algorithm. Furthermore, at the four marked positions in Figure 13, the presence of yellow data points indicates uninterrupted radar tracking, without the track dropout being observed. Thus, in terms of target track length, the ATPFC-IMM algorithm has a better performance than the CIMM algorithm and demonstrates superior overall tracking continuity.

## 6. Conclusions

In this paper, an adaptive maneuvering target tracking algorithm (ATPFC-IMM) applicable to a PD radar was proposed. The ATPFC-IMM algorithm combines the method of the adaptive correction of the probability transfer matrix and fuzzy control system. Especially, the fuzzy control system was designed to optimize the allocation of probabilities to each model. By employing two such approaches, the response time of model switching was effectively decreased, and the tracking performance of the ATPFC-IMM algorithm for the maneuvering target was enhanced. A comprehensive set of simulation and application analyses was conducted, compared with the classical IMM algorithm. The tracking accuracy and continuity of maneuvering target tracking were improved by the ATPFC-IMM algorithm. In the applications of specific PD radars, the ATPFC-IMM algorithm achieved a considerably satisfactory performance in maneuvering target tracking. In addition, the ATPFC-IMM algorithm exhibited sensitivity to measurement noise, as evidenced by the fluctuation in model probabilities within a relatively small range; but, it could still improve the tracking performance. In our future work, it is necessary to investigate strategies for noise resistance of ATPFC-IMM algorithms and improve the adaptive tracking performance of the algorithm when facing stronger maneuvering targets.

**Author Contributions:** Conceptualization, W.X. and J.X.; methodology, W.X.; software, J.X.; validation, D.X. and J.C.; formal analysis, H.W.; writing—original draft preparation, W.X. and J.X.; writing—review and editing, H.W. and J.C.; supervision, D.X. and H.W. All authors have read and agreed to the published version of the manuscript.

**Funding:** This research received no external funding.

**Data Availability Statement:** Data sharing not applicable. The simulation data for Experiment 1 and Experiment 2 can be obtained based on the simulation conditions. Due to privacy and other constraints, the application data are not suitable for sharing.

**Conflicts of Interest:** The authors declare no conflicts of interest.

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
