# Peer review of "An Adaptive IMM Algorithm for a PD Radar with Improved Maneuvering Target Tracking Performance"

_remotesensing, doi:10.3390/rs16061051_

Round 1

Reviewer 1 Report

Comments and Suggestions for Authors

This manuscript introduces an adaptive IMM algorithm with improved model switching speed and maneuvering target tracking accuracy. The authors also validate the value of the proposed algorithm in the application of PD radar in maneuvering target tracking through specific examples. However, some questions should be further clarified and revised. My detailed comments are outlined below.

1.    In the simulation experiments of this manuscript, the proposed ATPFC-IMM algorithm will be compared with the algorithm in reference [3]. However, there is a threshold setting for parallel algorithm switching in reference [3], and this article does not seem to explain the relevant threshold configuration. The threshold used in this paper should be explained.

2.   In the pseudocode of the ATPFC-IMM algorithm proposed in the manuscript, model probabilities is updated by circulation through steps 5 and 6. What are the conditions for the circular execution, and what are the conditions for the circulation to jump out or terminate? The author should explain this in the paper.

3.    In Section 5 of this paper, the authors first introduce some important parameters of the system application and the radar setup scenarios, but the lack of a flowchart of the system application processing or system design solutions of the proposed algorithm does not allow the reader to have a good understanding of the processing of the ATPFC-IMM algorithm in specific radar target tracking.

4.    In the references of the manuscript, some of the documents may be cited in an incorrect format, so the authors are requested to check them and revise them accordingly.

Comments on the Quality of English Language

English usage should be improved.

Reviewer 2 Report

Comments and Suggestions for Authors

The problem of estimating the state vector from observations in a multi-model formulation is considered, when an adaptive algorithm switches to one or another model depending on the observations.

In comparison with the known schemes, the authors propose an algorithm for switching between models based on the analysis of innovation processes after Kalman filters. The elements of the Markov transition matrix are also adaptively adjusted.

The tasks close to the considered formulation are described, for example, in the following articles:

X. Liu, F. Long, W. Zhang, Lu Guo. Modular Interacting Multiple Model Based on Extended Viterbi Algorithm for Maneuvering Target Tracking. Mathematical Problems in Engineering. Volume 2015, Article ID 374054, 6 pages

http://dx.doi.org/10.1155/2015/374054

G. Yuan, W. Zhu, W. Wang, B. Yin. Maneuvering Target Tracking Algorithm Based on Interacting Multiple Models Mathematical Problems in Engineering.  Volume 2015, Article ID 810613, 7 pages

http://dx.doi.org/10.1155/2015/810613

Le Minh Hoang, Aleksandr A. Konovalov, Dao Van Luc. Tracking of Maneuvering Targets Using a Variable Structure Multiple Model Algorithm. Journal of the Russian Universities. Radioelectronics. 2023, vol. 26, no. 3, pp. 77–89

The title of the article contains an indication of the use of an algorithm for a pulse radar, however, the task of tracking a target with a coordinated turn by a pulse locator can be solved without using a multi-model approach, and in some cases more effectively (see the sources given).

The article describes another possible variant of the multi-model approach, including additional adaptation. At the same time, it is important to identify the conditions and limitations under which additional adaptation is necessary. It is possible that in other cases it will be harmful and will only lead to a deterioration in the effectiveness of tracking.

The practical application of an adaptive multi-model approach to the chosen narrow task of determining the trajectory of a maneuvering target based on estimates of its coordinates by a pulse locator is questionable, since the article does not compare with the results already published in 2014-2015 on this issue.

It is also not very convincing to evaluate the effectiveness of tracking a target only with a standard error averaged over time, since the error is non-stationary in nature, and there are probabilities of large values and abnormal errors that can lead to a failure of tracking.

Reviewer 3 Report

Comments and Suggestions for Authors

The article presents an adaptive interacting multiple model (IMM) algorithm for tracking of a maneuvering target using pulse-Doppler (PD) radar. The IMM algorithm is made to be adaptive by correcting the probability transfer matrix in real time. Additionally, a fuzzy control system is utilized to update the model probability gain. The article concludes with some simulated and real-world examples followed by a short discussion.

Overall, the article presents an interesting method for target tracking using PD radar. However, it is not clear what the novel contributions of the paper are. Rather contributions to theoretical research, the article appears to be more focused on an engineering approach of conglomerating existing methods. While the latter approach is also welcome, there should be a stronger focus on comparisons with other methods and several more examples, both simulated and real-world.

Please find additional comments below, in no particular order:

1. The introduction is lacking. The existing methods that were cited do not have enough background and their benefits and drawbacks are not fully considered. For example, the article states "due to the addition of parallel algorithms, the complexity of the ATPM-PIMM algorithm has significantly increased". In that example, the complexity is not quantified. Additionally, in modern computational devices, even low powered devices such as an Nvidia Jetson Nano, parallel computations are possible which would make parallel algorithms beneficial. In general, there does not seem to be strong motivation for the proposed method.

2. The contributions of the article are not clear to the reader. It is advisable that the authors provide a clear list of contributions. Oftentimes a list of bulletpoints outlining the contributions provides a clear presentation of contributions.

3. The last paragraph of Section 2 does not provide strong or clear motivation as to why the classical IMM algorithm is a good candidate for optimizing the algorithm structure.

4. Running many different simulations and then taking a statistical approach to the results would provide the reader with more confidence in the method.

Comments on the Quality of English Language

Significant issues with English grammar, spelling, and punctuation. Errors make the article challenging to read and significantly distract the reader from the contributions.

Reviewer 4 Report

Comments and Suggestions for Authors

This paper proposes a novel target tracking algorithm for pulse-Doppler radar based on the interacting multiple model (IMM). The proposal sounds somehow interesting while their explanation and experimental results requires multiple points to be modified.

The authors have proposed a fuzzy algorithm to achieve an adaptiveness for the IMM while there have been multiple potential methods to gain adaptivity. It is not clear why and how the proposed strategy has an advantage compared to the other approaches. In addition, the descriptions in Sec. 3.2 are too short and simple. It must be more in detail including equations.

Some descriptions are redundant. For example, the authors showed a photo of UAV in Fig. 9 while it is nothing but a common drone without any modification. It must be better to show and summarize the property of the drone instead. For example, a Doppler radar requires specific speed for the target and thus, showing that the target fulfills the requirement.

Comments on the Quality of English Language

N/A

Round 2

Reviewer 2 Report

Comments and Suggestions for Authors

The authors did not respond to my comments regarding the results already obtained on the problem under study...I think they will take them into account in their future work.